# Superparamagnetic-like Micrometric Single Crystalline Magnetite for Biomedical Application Synthesis and Characterization

**Marius Chirita** [1,*], **Adrian Bezergheanu** [2], **Corneliu Bazil Cizmas** [2] and **Aurel Ercuta** [3]

1   National Institute for Research and Development in Electrochemistry and Condensed Matter, 300569 Timisoara, Romania
2   Transilvania University of Brasov, 500036 Brasov, Romania
3   West University of Timisoara, 300223 Timisoara, Romania
*   Correspondence: chirifiz@gmail.com

**Abstract:** Single-crystalline magnetite ($Fe_3O_4$) particles having a size beyond the nanometric range (1 μm to 50 μm) and showing high (close to the bulk value) saturation-specific magnetization ($\sigma_s$ = 92 emu/g), were obtained by the hydrothermal decomposition of the Fe-EDTA complex. The very low values of the magnetic remanence ($\sigma_r$ = 0.82 emu/g) and coercivity ($\mu_o H_c$ = 1.53 mT) observed at room temperature (RT) suggest a superparamagnetic-like behavior, which is quite remarkable for such micrometric magnetite particles. As confirmed by vibrating sample magnetometer (VSM)-based measurements, minor changes in their magnetic properties occur between RT and 5K. Scanning electron microscopy (SEM) has revealed a morphology consisting of a combination of non-porous octahedral- and dodecahedral-shaped particles, energy dispersive X-ray analysis (EDX) has indicated high elemental (Fe and O) purity, whereas X-ray diffraction (XRD) has confirmed a single crystal structure. The nitrogen adsorbtion–desorption isotherm and pore size distribution are presented for the magnetite sample. Thermomagnetic records under zero field-cooled (ZFC) and field-cooled (FC) conditions have revealed a thermal hysteresis of the Verwey transition.The Verwey point ($T_V$) at which the major step of the phase transformation takes place is located around 132 K for heating and around 122 K for cooling. These microcrystals do not remain agglomerated when the polarizing field is removed, an essential requirement in biomedical applications is met.

**Keywords:** single crystal; superparamagnetic; micrometric magnetite; Verwey transition

## 1. Introduction

It is known [1] that in magnetite ($Fe_3O_4$) particles of a size below10 nm, superparamagnetic relaxation occurs as a consequence of the thermally activated fluctuationsof the spontaneous magnetization, from one of the easy axes (the <111> crystal directions in this case) to another. With such a property (not often observed beyond the 10 nm–20 nm range), an important requirement in biomedical applications is met, i.e., the particles do not remain agglomerated when the magnetic field is removed. However, the very small size of these particles still causes two major drawbacks to their use in biomedical applications. On the one hand, some risks of both toxicity and the occurrence of physiological barriers for an enhanced permeability and retention (EPR) effect exist.These particles penetrate capillaries and strongly interact with the immune system [2]. On the other hand, due to the larger fraction of metal ions located in the external shell (usually affected by the spin disorder), the saturation magnetizations are significantly lower (commonly in the 30 emu/g–50 emu/g range [3,4]). As a solution to these drawbacks, among others, Bean and Livingston [5] have shown that superparamagnetic aggregates may easily be built from such nanoparticles, and in this sense, Silva et al. [6] reported synthesis by co-precipitation of micrometric agglomerations of $Fe_3O_4$ nanoparticles with a superparamagnetic core (11.8 μm) and amoxicillin

cover. Hinds et al. [7] showed that microparticles built from nanoparticle agglomeration could contribute to the improvement of the magnetic resonance imaging (MRI) signal, and also that these particles may effectively be endocytosed by various cells.the authors used these microparticles in the treatment of the gram-negative spiral-shaped Helicobacter pylori bacteria. Mankia and co-workers [8] have reported that mainly due to their physical size, microparticles of iron oxide create potent hypointense contrast effects in molecular MRI, and they used iron microparticles of size in the range of 1 μm as contrast agents in mouse brain inflammatory pathology for invivodetection of the disease. Additionally, they have mentioned that doses of micrometric iron oxide (MPIO) used in vivo animal studies have neither caused thrombosis, tissue infarction, or vessel plugging, nor any other noxious effects; moreover, their studies indicate that liver and spleen are faster at clearing the MPIO from the blood circulation than the ultrasmall particles. Many other works (e.g., [9–13]) refer to the degree of usefulness, especially for MRI applications, of such superparamagnetic microparticles built from nanoparticle agglomerations. However, a significant drawback remains, i.e., the saturation magnetization cannot exceed that of the constituent nanoparticles, which means that the magnetic response remains weak, with all the as-arising disadvantages, including the difficulty of handling and, more important, weak MRI response. Related to this issue, we have previously pointed out [14] that single crystalline microparticles of iron oxide with unusual superparamagnetic-like behavior (SCMSPIO) can be obtained by the hydrothermal decomposition of the $Na_4$-FeEDTA complex. We also have proved [15] that these particles can be used in MRI imaging and are biocompatible [16].

As we have already described the actual synthesis method in earlier articles, we shall only briefly discuss it here. However, we shall add details regarding how the high-pressure treatment times and the degrees of autoclave filling influence the synthesis result, i.e., final $Fe_3O_4$ crystals without any traces of hematite ($Fe_2O_3$) or siderite ($FeCO_3$). Additionally, regarding the as-obtained microcrystal characterization, apart from imagistic (SEM), elemental (EDX), and structural (XRD) analysis, we performed magnetic and thermomagnetic measurements. "Zero field-cooled" (ZFC) and "field-cooled" (FC) magnetization vs. temperature (in our case between 300 K and 5 K), was recorded, which clearly outlined the Verwey transition, a solid-state phase transformation(first observed in mineral magnetite), which is associated with changes in magnetic, electrical, and thermal properties [17–21]. Furthermore, we present the isothermal DC hysteresis loops, as the field dependence(H) of the specific (mass) magnetization was recorded within the 5 K–300 K thermal range. For the sake of simplicity, in [16], we generically labeled these particles as "Single-Crystalline Micrometric Superparamagnetic Iron Oxide" (SCMSPIO), an abbreviation that we shall also use in the present paper.

## 2. Materials and Methods

### 2.1. Materials and SCMPIO Synthesis

Ferric ammonium sulphate$FeNH_4(SO_4)_2 \bullet 12H_2O$ (FAS), tetrasodium ethylenediaminetetraacetate ($Na_4EDTA$), urea ($NH_2)_2CO$, all of analytical purity, were supplied by Fluka (Sigma-Aldrich, St. Louis, MO, USA).

The "in two steps" method was used to synthesize SCMSPIO.

- First, the Fe-EDTA complex was obtained in a $1.05 \times 10^{-1}$ M solution of FAS, $1.05 \times 10^{-1}$ M solution of $Na_4EDTA$ and a $9.71 \times 10^{-1}$ M urea solution (all aqueous); the Fe(III)-EDTA complex formation is indicated by the dark red color.
- Second, the hydrothermal method was used for $[Fe(III)EDTA]^- + Na^+$ decomposition, for which the Fe-EDTA solution was transferred into a 70 mL Teflonstainless steel line autoclave and was heated up to 230 °C, at a rate of 1.7 °C/min. The degree of filling was 50%. After 30 h of high-temperature treatment, the autoclave was abruptly water-cooled to ensure the freezing of any ongoing processes; inside the autoclave, the pH of the solution was 9.4. After washing in bidistilled water, a fine powder of shining black particles was obtained.

### 2.2. SCMPIO Characterization

A Quanta 3D 200i, FEI Co. (Hillsboro, OR, USA), scanning electron microscope (SEM) with an energy dispersive X-ray (EDX) system was used for morphological examination and elemental analysis.

Crystal structure analysis was performed at room temperature (RT) using the X'Pert PRO MPD (PANalytical) diffractometer, using Cu-K$_\alpha$ radiation (0.15418 nm) in a θ:2θ configuration with a monochromator. A Quantachrome NOVA 1200e device was used to record the adsorption–desorption isotherm. Magnetic measurements were performed with accuracy and a reproducibility rate of 0.5%, and a noise base of $10^{-6}$ emu using the VSM (Vibrating Sample Magnetometer) facility of the 7T Mini Cryogen Free Measurement System (Cryogenic Ltd., London, UK). This equipment is an automatic cryogenic system with a closed helium circuit for measuring magnetic, electrical, and thermal properties of materials in applied DC magnetic fields from 0 T to 7 T at low temperatures from 1.6 K to 325 K, with a high-temperature option up to 700 K for magnetic measurements. All magnetic measurements were performed on non-orientated powders so the orientation of the magnetic field with respect to the microparticles was arbitrary.

## 3. Results and Discussion

### 3.1. Morphology

The SEM images in Figure 1 unequivocally show well-dispersed $Fe_3O_4$ microparticles; this indicates a lack (or at least a very low value) of magnetic remanence.

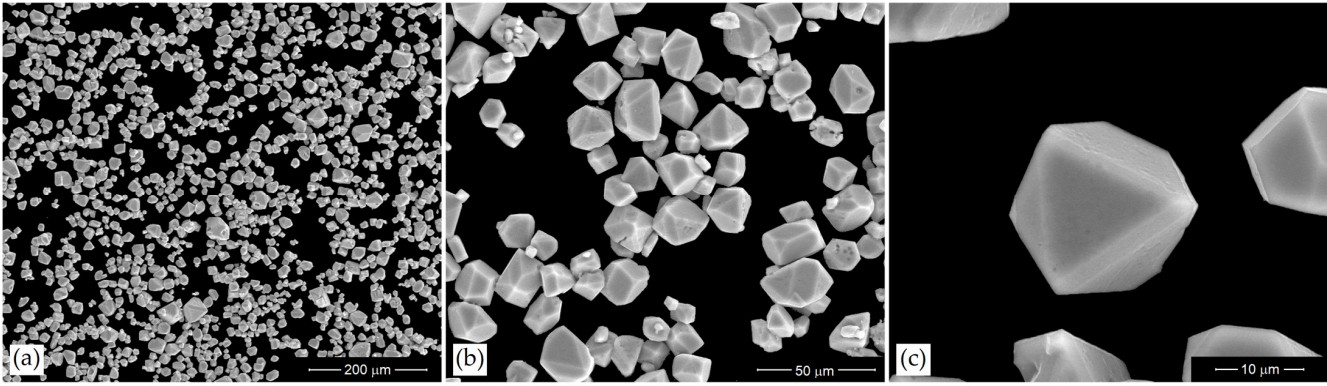

**Figure 1.** SEM images of the $Fe_3O_4$ dispersed powder, including (**a**) 400× magnification, (**b**) 1600× magnification, and (**c**) 6000× magnification with morphology details.

It can be seen that the microcrystal morphology is a combination of octahedral and dodecahedral faces. In addition, a compact habit structure without porosity is observed.

### 3.2. Elemental Analysis

In Figure 2, the EDX spectrum of the 30 h final reaction product is shown; here, no impurities on the surface of the microcrystals are confirmed by the absence of any Na, S, C, or N traces that could possibly be remaining from EDTA and FAS decomposition.

### 3.3. Crystal Structure

From the XRD record shown in Figure 3, the symmetry of the crystallites was identified to belong to the Inorganic Crystal Structure Database (ICSD), reference code 01-075-0033, pattern for $Fe_3O_4$.

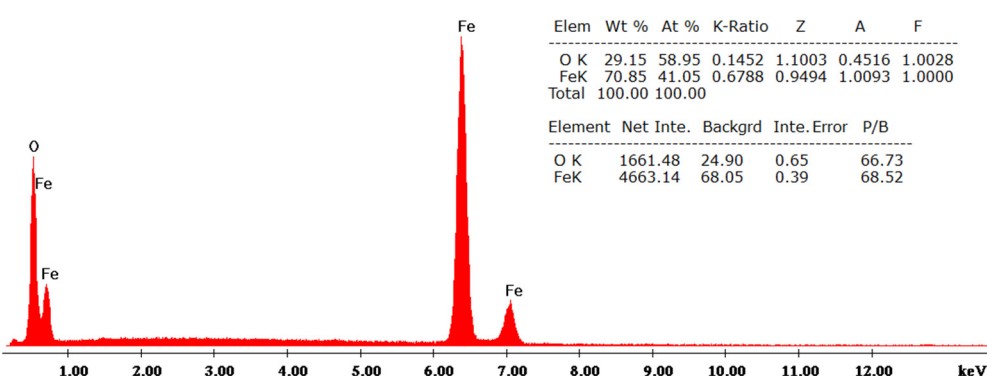

**Figure 2.** EDX spectrum of the magnetite powder.

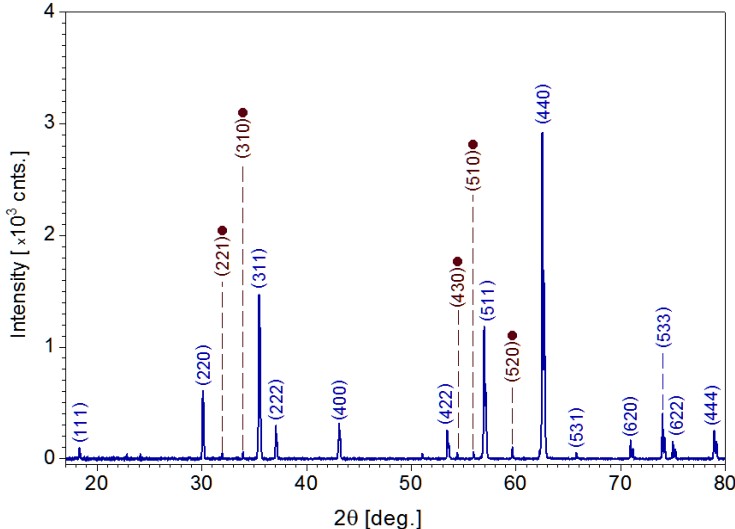

**Figure 3.** The XRD record of the microcrystal powder.

However, the small diffraction peaks (marked by brown bullets) observed at 2θ = 32.125 hkl [221], 2θ = 33.883 hkl [310], 2θ = 55.56 hkl [430], 2θ = 56.107 hkl [510], and 2θ = 59.570 hkl [520] were found to match the ICSD, reference code 00-039-1346, pattern for ϒ-Fe$_2$O$_3$; this fact is consistent with surface oxidation of some crystallites.

### 3.4. Specific Surface and Porosity

Figure 4 reveals the nitrogen adsorption–desorption isotherm of the magnetite sample and the pore size distribution. Before we analyze the sample, the material was degassed in a vacuum overnight, at room temperature.

Analyzing the adsorption–desorption isotherm obtained and compared with IUPAC references [22], we conclude that the sorption isotherms are type II, specific for non-porous materials. The textural parameters were determined using computational methods. The specific surface area was determined using the Brunauer, Emmet, Teller (BET) method, indicating a value of 5.8 m$^2$/g. The total pore volume calculated from the last point of isotherm indicates a value of 0.012 cm$^3$/g. When applying the density functional theory (DFT) method, an average pore size of approximately 2 nm for Fe$_3$O$_4$ crystallites was obtained. The rugosity value(D)of 2.0861,as determined by the Frenkel, Halsey, and Hill (FHH) method, indicates that the material tends to be in a 2D state with a smooth surface [23].

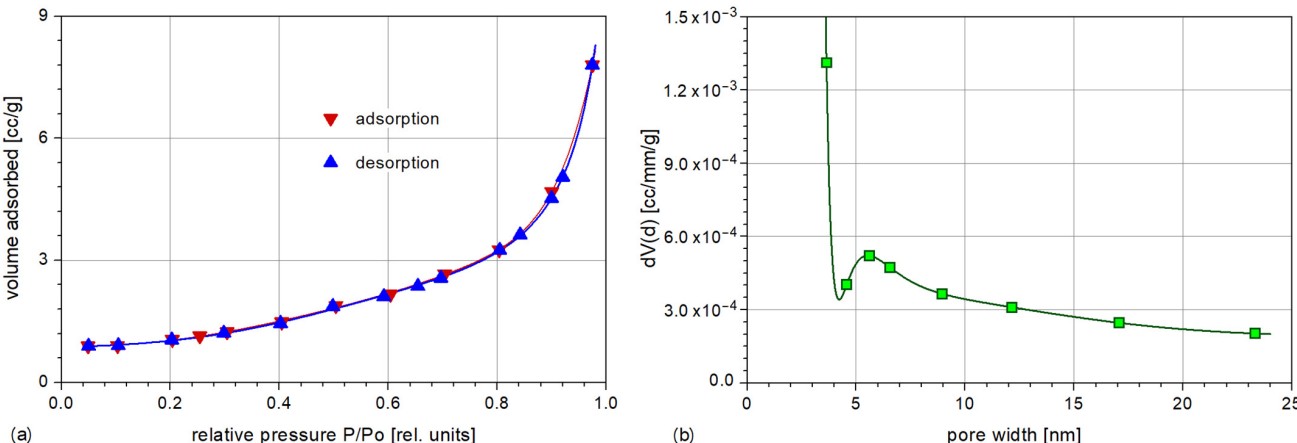

**Figure 4.** (**a**) The nitrogen adsorption–desorption isotherm of the magnetite sample and (**b**) pore size distribution.

### 3.5. Magnetic Properties

The isothermal DC hysteresis loops, plotted in Figure 5 as the field dependence $\sigma(H)$ of the specific (mass) magnetization, were recorded within the 5 K–300 K thermal range. If the coercive force and magnetic remanence values are extracted from thisloop and plotted as their temperature dependence (Figure 6), one can see that above 150 K the magnetization cycle of the cubic inverse spinel phase is effectively anhysteretic. From this point of view, the magnetic behavior is superparamagnetic-like.

In addition, the differences in the ease of magnetization of magnetite crystals in the two structural phases (monoclinic and cubic), first reported by Ozdemir [24], are confirmed. These occur between the $\sigma(H)$ slope in the low field region and between the curvatures of the "mid-field" region "knee" of this dependence [25,26]. This knee region joins the field range dominated by the irreversible domain wall motion (when the applied field strength is increased) or by nucleation and growth of inverse domains (when the field is decreased) and the higher field range dominated by magnetization rotations.

In order to clarify the thermal evolution of the crystal structure, specific magnetization vs. temperature records were performed using the so-called "zero field cooled-field cooled" (ZFC-FC) method. To this end, the sample was first cooled from 300 K down to 5 K, and then a low DC field ($\mu_o H = 10$ mT) was applied and maintained constant during a complete heating/cooling cycle between the above thermal limits; the plot of these records is shown in Figure 7. Here, the Verwey phase transformation, from cubic to a lower symmetry on cooling and in reverse order on heating, may be observed.A debate exists in the literature on whether the low-temperature phase is monoclinic, pseudo monoclinic, or rhombohedral [27].

As the temperature passes across the Verwey point, a steep jump of the magnetization occurs at $T_{Vh} = 132$ K on heating and at $T_{Vc} = 122$ K on cooling, which means that the Verwey transition inmagnetite exhibits thermal hysteresis, an almost unnoticed phenomenon in magnetite but often observed in the solid-state reversible phase transformations (less those involving atoms diffusion); in this sense, the martensitic transformation in shape memory alloys [28] is one of the most studied.

It is worth noting that, unlike systems of non-interacting single-domain nanoparticles (magnetostatic interaction), powders from multi-domain microcrystals are not *stricto-sensu* superparamagnetic, even if these powders exhibit very small $\sigma_R$ and $H_C$. Indeed, apart from this criterion for superparamagnetism (anhysteretic magnetization cycle), additional requirements are expected to be met.

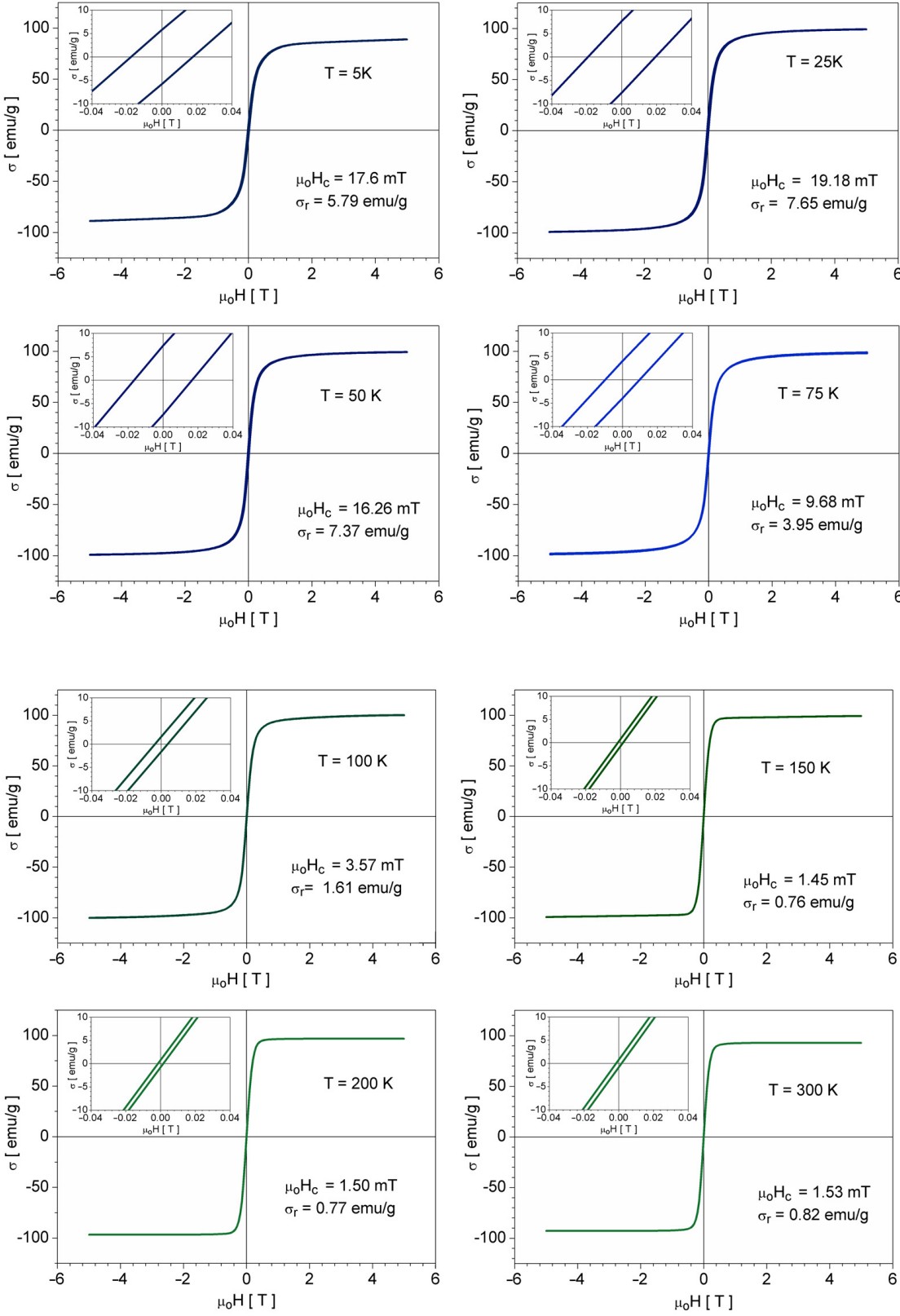

**Figure 5.** Magnetic hysteresis loops are recorded at temperatures between 5 K and 300 K.

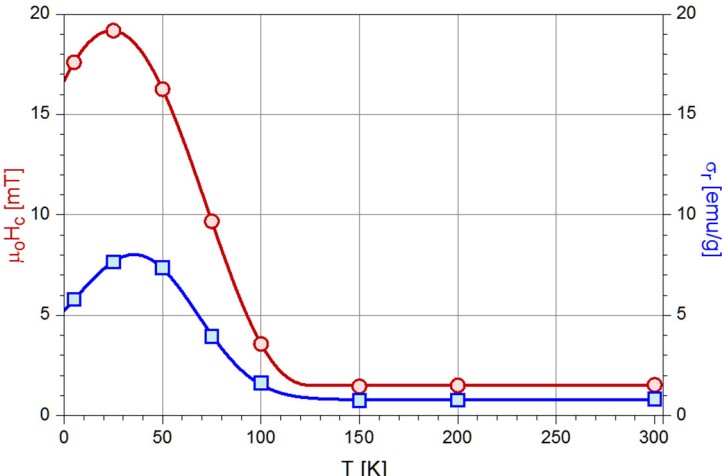

**Figure 6.** Thermal evolution between 5 K and 300 K of the DC coercive field ($\mu_o H_c$) and the magnetite microcrystals' remanent magnetization ($\sigma_r$).

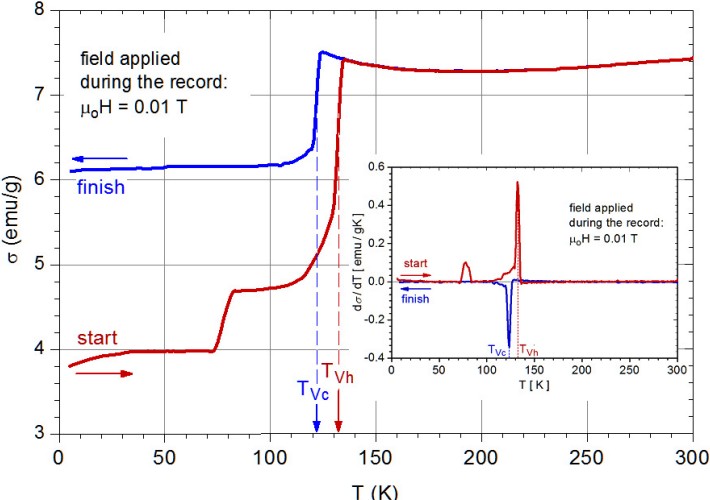

**Figure 7.** ZFC-FC magnetization vs. temperature records of the magnetite microcrystals. In the inset, the magnetization rate of change $d\sigma/dT$ is plotted.

- The coincidence of the isothermal magnetization curves (or cycles) when plotted against H/T (instead of H), a direct consequence of the fact that the profile of the $\sigma$(H) dependence of an assembly of monodisperse superparamagnetic particles, is described by a Langevin function:

$$L(\xi) = \coth(\xi) - \frac{1}{\xi} \qquad \text{with} \qquad \xi = \frac{\mu_o m_p H}{k_B T} \qquad (1)$$

where $\mu_o$ is the vacuum permeability, $k_B$ is Boltzman's constant, and $m_p$ is the magnetic moment of a particle, or by a superposition of such functions in the case of a polydisperse assembly ($m_p$ depends on the particle volume).

- The specific profile of the ZFC thermomagnetic record, i.e., the presence of a maximum at the blocking temperature ($T_B$), which for a given compositionis directly related to the nanoparticle size.

Clearly, neither of these last two conditions holds for microcrystalline powders.

### 3.6. Hydrothermal Synthesis and Chemical Reactions

In order to obtain single-phase magnetite, after many attempts presented in [14,16,29], the high-pressure treatment time and the temperature were set to 30 h and 230 °C, respectively. We mention that the autoclaves were initially filled to 70% capacity. However, one of the main drawbacks was the random presence (only sometimes) in the autoclave of small or very small amounts of iron carbonate ($FeCO_3$). Because the two phases cannot be perfectly separated, initially, we found a method to remove $FeCO_3$ crystallites quite easily by dissolving (three hours, a solution of HCl, and pH = 0.5, 70 °C), minimizing the risk of the Fe ions leakage from the magnetite structure or the properties of the magnetite microcrystals. Then, to avoid unwanted $FeCO_3$ formation, we tested different degrees of filling the autoclave. Without changing the concentrations of the precursors, we used several autoclaves with different degrees of filling and found that a degree of filling of 50% or less ensures the presence of magnetite in the single phase after 30 to 40 h of high-pressure treatment. Complete control of the particle size, progressively growing from 1 μm to 40 μm, is thus achieved, and no acid (HCl)- based washing was needed. The complete evolution of the mixture of $Fe_3O_4$ and $FeCO_3$ phases and how this evolution depends on the degree of the autoclave filling and high-pressure treatment times are presented in [16].

Regarding chemistry, the synthesis starts from Fe(III) ions and includes the use of reducing agents. As EDTA was introduced into the FAS solution, the color of the solution's color turned red, an indicator that an Fe(III)EDTA complex was formed. The dominant compound in the solution depends on its pH [30]. At pH = 3, the formation of $[FeIII(OH_2)EDTA]^-$ takes place, whereas the $[FeIII(OH)EDTA]^{2-}$ complex becomes dominant when the pH value increases to 5; the following reaction:

$$2\left[FeIII(OH)EDTA\right]^{2-} \leftrightarrow \left[EDTAFeIII - O - Fe^{III}EDTA\right]^{4-} + H_2O \qquad (2)$$

takes place [26]. The red color of the solution is because the dimer species appears. Previous studies regarding the thermal behavior of the Fe(III)EDTA complex have shown that its decomposition temperature is 140 °C. When the temperature is 140 °C [28], the complex Fe(III)-EDTA decomposes by an internal electron transfer process [31]. As the temperature increases, EDTA is simultaneously involved in the redox with Fe(III) and hydrolysis reactions. From the latter reaction, IDA (iminodiacetic acid), and HEIDA (hydroxyethyliminodiacetic acid) species result in the following:

$$EDTA^{4-} + H_2O \rightarrow IDA^{2-} + HEIDA^{2-} \qquad (3)$$

Compared to EDTA, the chelating agents IDA and HEIDA are more stable at the process temperatures. If $Fe^{3+}$ ions exist in the solution, Fe(III)IDA and Fe(III)HEIDA complexes form at temperatures starting from 150 °C [28]. In an alkaline solution, the chelating agent decomposes into ED3A (ethylenediamonotriacetic acid) and $CO_3^{2-}$ (reaction 4). Therefore, $CO_3^{2-}$ ions are created by the hydrolysis of urea in conformity with reaction 4 [31]. In a previous study [29], we proved the existence of $FeCO_3$ crystals in pure form after high-pressure temperature treatment between 12 h and 22 h at 230 °C (reaction 6); after 26 h at 230 °C, magnetite and ferrous carbonate coexist. Magnetite is synthesized in an alkaline medium at a high-pressure temperature treatment time, longer than 28 h at 230 °C by the reaction of Fe (III) ions, resulting from the hydrolysis of Fe (III) complexes and $Fe^{2+}$ ions from $FeCO_{3(s)}$. The following reactions successively take place:

$$[Fe(III)EDTA]^- + 3HO^- + Fe^{3+} \rightarrow ED3A^{3-} + CO_3^{2-} + 2Fe^{2+} + CH_2O + H_2O \qquad (4)$$

$$NH_2(CO)NH_2 + 2H_2O \rightarrow 2NH_4^+ + CO_3^{2-} \qquad (5)$$

$$Fe^{2+} + CO_3^{2-} \leftrightarrow FeCO_{3(s)} \qquad (6)$$

$$2Fe^{3+} + FeCO_{3(s)} + 8HO^- \rightarrow Fe_3O_4 + CO_3^{2-} + 4H_2O \qquad (7)$$

## 4. Conclusions

Superparamagnetic-like SCMSPIO particles with sizes ranging from 1 μm to 40 μm showing $\sigma_s$ = 92 emu/g for saturation-specific magnetization, $\sigma_r$ = 0.82 emu/g for magnetic remanence, and $\mu_o H_c$ = 1.53 mT for coercivity at RT, were obtained by the hydrothermal decomposition of the Fe-EDTA complex. Such behavior is very unusual for single-crystalline particles having micrometric sizes which should rather be ferromagnetic-like. These magnetic qualities are appropriate for biomedical applications, due to the fact that the particles do not remain agglomerated after the removal of the magnetic field. SCMSPIO could be useful in some biomedical applications where the nanoparticles have major drawbacks (as we described above). Selecting their sizes and applications, such as monitoring cell migration for cell therapy, cellular MRI, MRI contrast agents, cell labeling, detection, immobilization, modification of biologically active compounds, magnetic separation of cells, and others, could benefit from using SCMSPIO. Regarding the characterization, minor changes in their magnetic properties occur between RT and 5 K. A thermal hysteresis of the Verwey transition occurs between 132 K on heating and around 122 K on cooling under ZFC and FC, respectively. In addition, these steps exhibit "tails" directed toward lower temperatures, extending down to ca. 80 K on heating and ca. 100 K on cooling. These tails are attributed to natural oxidation (either during the synthesis process or in the time interval between synthesis and XRD measurements) that has affected a limited number of crystal particles, and this is consistent with the weak maghemite ($Fe_2O_3$) peaks detected in the XRD record.

**Author Contributions:** M.C. conducted the synthesis of SCMSPIO and the SEM, XRD, and EDAX interpretation. A.B. and C.B.C. conducted the magnetic and thermomagnetic characterization. A.E. conducted the magnetic and thermomagnetic interpretations. All authors have made equal contributions to the writing of the manuscript. All authors have read and agreed to the published version of the manuscript.

**Funding:** This research received no external funding.

**Institutional Review Board Statement:** Not applicable.

**Informed Consent Statement:** Not applicable.

**Data Availability Statement:** Not applicable.

**Acknowledgments:** The authors thank Radu Banica, Catalin Ianasi, Daniel Ursu, Liviu Mocanu, and Corina Orha for technical support and helpful discussions.

**Conflicts of Interest:** The authors declare no conflict of interest.

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
