# Peer review of "Superparamagnetic-like Micrometric Single Crystalline Magnetite for Biomedical Application Synthesis and Characterization"

_magnetochemistry, doi:10.3390/magnetochemistry9010005_

Round 1
Reviewer 1 Report
A very interesting paper on microscopic magnetic materials. I have the following comments.
Figure one SEM images need a scale bar
Also needs a size distribution histogram
Did you determine crystalite size using Scherrer equation from XRD pattern in figure 3? This may indicate a possible reason for superparamagnetic behavior.
The central section of the VSM results should be magnified in order to better demonstrate the superparamagnetic behavior.
There are a couple of mistakes in the English but nothing too grave.
Author Response
Reviewer 1
Figure one SEM images need a scale bar
Response: The scale bars were introduced in figure 1.
- Also needs a size distribution histogram.
Response: Agree; a size histogram would bring additional information. However, a set of SEM images containing a sufficiently large number (at least several hundred, if not thousands) of non-overlapping particles is needed. By all means, Fig.1.a does not meet the requirements for a correct histogram. In addition, generating such a set of appropriate SEM images needs more time, which we apparently cannot rely. Of course, if you consider this task mandatory, and if Editor grants us the required time, we can accomplish this.
- Did you determine crystalite size using Scherer equation from XRD pattern in figure 3? This may indicate a possible reason for superparamagnetic behavior.
Response: Thanks for the suggestion. We thought about that too. We applied Sherer's formula often when it came to small particles (usually below 600 nm) that had peaks of the diffraction spectra with a large width. However, in our case, the diffraction maxima are very narrow and sharp which is a powerful indicator for large dimensions of crystallites. That is why the application of Sherer's formula would not bring anything new; especially at these dimensions (order of tens of microns), the formula loses its validity.
- The central section of the VSM results should be magnified in order to better demonstrate the superparamagnetic behavior.
Response: The central section of the VSM results was magnified –inset in fig 5.
- There are a couple of mistakes in the English but nothing too grave.
Response: The English was improved.

Reviewer 2 Report
The manuscript by Chirita et al presents the magnetic characterization of magnetite microparticles. The manuscript is well presented and clear. The main issue with this manuscript is it content. The preparation, functional characterization and biocompatibility of the microparticles have all been reported before by the authors as they mention in the draft (refs 14 to 16). Thus the report of exclusively the magnetic properties of these samples do not justify publication.
Apart from this, a couple of other more concrete points:
-Figure 1, SEM. Scale bars are missing. Also from these images the authors conclude that the particles are not porous. With the resolution provided by these images (probable in the micron range) it is not possible to discard the presence of pores.
-Figure 2, EDXS. The spectra looks unnaturally clean (not saying with this that it is fabricated). But the absence of other peaks beyond O ad Fe means that there is nothing on top of the particles, no coating whatsoever, not even impurities... no C or N from the starting materials...
-In section 2.1 the authors when describing the synthesis say that the reaction is kept at 230 ºC for 30h. In section 3.5, they say it is 32 h.
Overall the quality of the manuscript is good, but the content do not justify publication in my opinion.
Author Response
Reviewer 2
Comments and Suggestions for Authors
- Figure 1, SEM. Scale bars are missing. Also from these images the authors conclude that the particles are not porous. With the resolution provided by these images (probable in the micron range) it is not possible to discard the presence of pores.
Response: The scale bars were introduced in the SEM images. BET analysis was performed and added in subsection 3.4, figure 4. “Specific surface and porosity" was added to the manuscript. Based on adsorption-desorption records, we have concluded that porosity is practically absent.
- Figure 2, EDXS. The spectra looks unnaturally clean (not saying with this that it is fabricated). But the absence of other peaks beyond O ad Fe means that there is nothing on top of the particles, no coating whatever, not even impurities... no C or N from the starting materials...
Response: Indeed, no traces of any elements that should result from EDTA or FAS decomposition have appeared in the EDAX spectrum. We have many similar EDAX records (maybe a few dozen). In this manuscript, we replaced the old image with one in which the whole range of energies can be seen to certify that the surface is clean. A possible explanation for the “unnatural” fact that the RDX spectra look clean may be due to the lack of porosity and surface softness. Also, the final solution is slightly yellowish and perfectly transparent, which indicates no more reactant in the autoclave. No changes occur at longer autoclaving times (longer than 30 hours).
So, in the sense of your comments, in 3.2 ,“the high purity of the microcrystalites” was replace with “no impurities on the surface of the microcrystallites”.
- In section 2.1 the authors when describing the synthesis say that the reaction is kept at 230 ºC for 30h. In section 3.5, they say it is 32 h.
Response: in paragraph 3.6 (3.5 became 3.6), we changed the value to 30h.
- Overall the quality of the manuscript is good, but the content do not justify publication in my opinion.
Response: We disagree with that statement.
The quality of the microcrystals is clearly shown by SEM, purity, and lack of pores, as indicated by the BET analysis. These properties have not been discussed previously. Also, the closeness of the behavior of these microparticles to superparamagnetism is discussed in detail here. Not to mention that the Magnetochemistry journal is focused to a large extent just on magnetic properties and application on materials (usually synthesis products).

Reviewer 3 Report
Article reference: Magnetochemistry 2068268
Title : Superparamagnetic-like Micrometric Single Crystalline Magnetite for Biomedical Applications-Synthesis and Characterization
Authors : Marius Chirita, Adrian Bezergheanu, Corneliu Bazil Cizmas and Aurel Ercuta
In the present paper, the authors report experimental results on single crystalline micrometric magnetite (Fe3O4) particles. The structure and the morphology were investigated using diffraction (XRD) and Scanning Electron Microscopy (SEM); while energy dispersive X-ray analysis (EDX) was used for the chemical analysis of the samples. The magnetic properties were studied by means of the vibrating sample magnetometer (VSM) with temperature varying from RT down to 5K. The paper fits the scope of Magnetochemistry; Special Issue: Advances in Magnetic Nanocarrier for Biomedical Applications .
The following remarks can be made about this manuscript.
Some suggestions are given in the following to make the study more complete; some information can be added for the benefit of the reader.
1/ The particle sizes are in the 1 to 40 mm range, i.e. in the micrometric range. What are the differences and the similarities in the magnetic behaviours of such micrometric particles compared to the nanometric ones ?
- For instance, are the features shown in Figs. 5 (coercive field and remanent magnetization vs temperature) and 6 (magnetization vs temperature) observed also for nanometric particles ?
- How about the different magnetic parameter values, such as the coercive fields and the remnant magnetizations.
If the authors have such information, they could briefly made a comparison. May be the authors did discuss this issue in their previous work (refs. 14 and 16), but I believe, it will be interesting if they summarize the main points of the comparison between the two particle size ranges.
2/ The application of a magnetic field in the VSM experiment and the field cooled method.
It will be useful if the authors indicate the direction of the applied magnetic field. Is it arbitrary or along a specific direction, such as a given crystallographic direction, or is it along the easy direction of the magnetization... ?
About the presentation
Some minor errors and misprints should be corrected. For instance,
- In page 2 " inin vivo".
- In page 2, "first" and "second" should be "First" and "Second".
I suggest that the authors go throughout the manuscript and correct all the misprints.
For the references : The article titles of some references are missing (refs. 26, 27, 28 and 30).

Author Response
Reviewer 3.
Agree. To complete the study, some information (typed red) was added at the end of Subsect.3.4. Magnetic properties, and in this sense, your suggestions were considered.
- The particle sizes are in the 1 to 40 mm range, i.e. in the micrometric range.
What are the differences and the similarities in the magnetic behaviors of such micrometric particles compared to the nanometric ones?
- For instance, are the features shown in Figs. 5 (coercive field and remanent magnetization vs temperature) and 6 (magnetization vs temperature) observed also for nanometric particles ?
- How about the different magnetic parameter values, such as the coercive fields and the remnant magnetizations.
- If the authors have such information, they could briefly make a comparison. May be the authors did discuss this issue in their previous work (refs. 14 and 16), but I believe, it will be interesting if they summarize the main points of the comparison between the two particle size ranges.
Response: We have inserted comments at the end of paragraph 3.5.
In addition, we have some comments just for you:
Authors’ response
Since apart from chemical composition, structure defects (internal stresses, dislocations, vacancies, non-magnetic inclusions, etc.), and exert a strong influence on the magnetic properties of materials, especially if these materials exhibit magnetostriction, a comparison between the magnetic behaviors of the micrometric and nanometric particles is not an easy task, even if the two species of particleshave the same chemical composition. Moreover,the magnetic interactions between the particles (causing particles chains or agglomerations formation), especially if these are of nanometric size, are expected to play a certain role.
Yet, a simplified discussion restricted to powders frommagnetite particles of high chemical purity and crystallinity, may reveal several differences and similarities in their magnetic behaviour. We shall take into account the fact that above the Verwey temperature (TV≈120 K) magnetite exhibits inverse spinel cubic structure, and that the magnetic moments of the Fe ions (Fe2+ and Fe3+) lie along the <111> directions called “axes of easy magnetization” (or “easy axes”), forming two “sublattices”, one containing Fe3+ ions located in the sites with tetrahedral oxygen coordination (A), and the other containing a mixture of Fe3+ and Fe2+ ionsin equal proportions, located in the sites with octahedral oxygen coordination (B).The magnetic moments of the Fe3+ ions from the two sublattices lie mutually antiparallel, thus cancelling, a 4B magnetic moment per unit cell results(B is the Bohr magneton) as an exclusive contribution from theFe2+ ions; this arrangement corresponds to an “uncompensated antiferromagnetism”called “ferrimagnetism”. As the temperature decreases below TV, the crystal symmetry changes from inverse cubic spinel to monoclinic (the low temperature crystal symmetry is not yet fully elucidated, orthorhombic and direct ), and the easy axes change from the cubic <001> directions to the monoclinic c axis. This explains why a step-down change (on cooling), or a step-upchange (on heating)of the magnetization occurs under weak DC field conditions, as the temperature varies across TV (it is worth to note that the Verwey phase transformation in magnetite exhibits thermal hysteresis)
Differences.
- a) in what regards the evolution of magnetization vs.applied field,M=M(H), these take place by different mechanisms in the two particles size cases.
Thus:
a.1) to reduce magnetostatic energy (by closing the magnetic flux lines inside the material), magnetite microparticles, like the other ferromagnetic and ferrimagnetic bulk materials,are spontaneously divided into regions called “magnetic domains” (or “Weiss domains”), inside which the metal ions’ magnetic moments lie along the same easy axis; conventionally, these moments are represented by the their vectorial sum per unit volume (MS), called “spontaneous magnetization”. Since the MS orientations are different (either antiparallel along the same <111> direction, or along different<111> directions) in adjacent domains, a gradual rotation between the two moments orientations takes place within a transition region (extended from several tens of atomic layers in soft magnetic materials, to thousands of such layers in hard magnetic materials) called “domain wall” (“Bloch wall” in bulk materials, and “Neel wall” in thin films, the direction of the moments rotation axis being normal or parallel to the wall, respectively). In multidomain materials, the state of magnetization is changed by the applied field, in principal by three processes:
a.1.1) reversible domain wall motion in weak fields.
a.1.2) irreversible domain wall motion (called “Barkhausen jumps”) in moderate fields; as the field strength increases, the magnetic domain structure evolves towards the single domain state.
a.1.3) coherent rotations (reversible or irreversible) of the MS vector in strong fields.
As a consequence of the irreversible processes involved, the evolution of the overall magnetization state depends on the recent magnetic history, which in the case of increasing and decreasing applied field leads to different paths of the ascending and descending branches of the M(H) dependence;under cyclic fields, the profile of this dependence is a magnetic hysteresis loop. From the major hysteresis loop, along which the “up” and “down” states of magnetic saturation (technical saturation) are successively reached, the saturation and remanent magnetization, Msat, and MR, respectively, as well as the coercive field (sometimes called“coercive force”), HC, are currently determined. While Msat is a specific material parameter (note that for in homogeneous systems like powders, ferrofluids, composites, etc., the specific magnetization =M/, where is the mass density, is preferred ), both MR, and HC may vary within significant limits for the same material, depending on the stress state, and even on particles’ size and shape. In this sense, the very close to anhysteretic shape (very low MR, and HC) of the cyclic magnetization loop exhibited by our magnetite microcrystals is quite remarkable.
a.2) in assemblies of nanoparticles, which are single domain particles below a critical size (approx. 27 nm for magnetite), the overall magnetization changes take place by:
a.2.1) particles rotation towards the field direction (Brown processes),when these particles are free to move (e.g. like in ferrofluids at temperatures above the freezing point of the liquid matrix).
a.2.2) magnetization rotation inside the particles (Neel processes), when these particles are immobilized (e.g. by forming aggregates, or being blocked in a frozen fluid or in a solid matrix).
The time rates of these two processes are temperature-dependent, and the associated relaxation times
are temperature (T) dependent. Here, kB is the Boltzman constant, is the matrix viscosity,Vh is the hydrodynamic volume of the particle,0~10-11-10-9s is an “attempt time”, K is the first anisotropy constant, and VM is the magnetic volume of the particle (see, e.g. T.E . Torres, E. Lima Jr., M. Calatayud, B. Sanz, A. Ibarra, R. Fernández-Pacheco, A. Mayoral, C. Marquin, M.R. Ibarra, and G.F. Goya-The relevance of Brownian relaxation as power absorption mechanism in Magnetic Hyperthermia, Scientific Reports (2019) 9:3992, https://doi.org/10.1038/s41598-019-40341-y).
When the two processes are simultaneously active, an effective relaxation time:
is considered. Both B, and Nare significantly shorterthan the time window (tm) required for measuring a single data point on the DC or low frequency AC magnetization cycle(note that generation at macroscopic scale of strong AC fields required to drive the particles assembly to magnetic saturationof frequency exceeding several hundreds of Hz,is a hard technical task.
- b) In what regards the thermomagnetic behaviour, the ZFC ad FC curves recorded for microparticles and nanoparticles powders exhibit completely different profiles:
- whereas the ZFC and FC thermograms records clearly highlight the Verwey transition with thermal hysteresis in the case of the micrometric particles, the phase transition is hardly visible (or even invisible) in the case of the latter; instead, the freezing of the magnetic moments until the temperature reaches TB (the blocking temperature) on heating after ZFC; two examples are given below:
(also see: M. Knobel, W. C. Nunes, L. M. Socolovsky, E. De Biasi, J. M. Vargas, and J. C. Denardin,Superparamagnetism and Other Magnetic Features in Granular Materials; A Review on Ideal and Real Systems, Journal of Nanoscience and Nanotechnology, Vol.8, 2836–2857, 2008
Similarities.
-some intrinsic properties, like magnetocrystalline anisotropy, magnetostriction, and Neel temperature, are considered as the same, regardless the size range
Reviewer’s comment #2)
- If the authors have such information, they could briefly made a comparison. May be the authors did discuss this issue in their previous work (refs. 14 and 16), but I believe, it will be interesting if they summarize the main points of the comparison between the two particle size ranges.
Authors’ response
Such comparison was discussed in refs. 14 and 16.
2) The application of a magnetic field in the VSM experiment and the field cooled method.
It will be useful if the authors indicate the direction of the applied magnetic field. Is it arbitrary or along a specific direction, such as a given crystallographic direction, or is it along the easy direction of the magnetization... ?
Response: “All magnetic measurements were performed on non-orientated powders, so the orientation of the magnetic field with respect to the microparticles was arbitrary.” This statement was added at the final of the 2.2.
About the presentation:
Some minor errors and misprints should be corrected. For instance,
- In page 2 " inin vivo".
Response: The change has been made.
- In page 2, "first" and "second" should be "First" and "Second".
Response: The change has been made.
I suggest that the authors go throughout the manuscript and correct all the misprints.
Response: The changes have been made.
For the references: The article titles of some references are missing (refs. 26, 27, 28 and 30).
Response: The changes have been made….paper 28 was eliminated. Works 23 and 24 were added.

Reviewer 4 Report
Manuscript: Superparamagnetic-like Micrometric Single Crystalline Magnetite for Biomedical Applications- Synthesis and Characterization
In presented study authors investigated magnetic properties of Single-Crystalline Micrometric Superparamagnetic Iron Oxide (SCMSPIO) synthesized by thermal decomposition of Fe3+ complex with EDTA to improve its biomedical usage. The idea is very interesting and can attract scientific audience. However, there are some corrections authors should made before article can be accepted for the publication.
1. Replace singlecrystalline with single-crystalline.
2. The formula Fe(III)-Na4EDTA is incorrect since iron(III) replaces 3Na+ cations from tetrasodium ethylenediaminetetraacetic acid, and so the name Iron-III-Tetrasodium-Ethylenediaminetetraacetic Acid complex is also incorrect. Please, correct it.
3. Line 16: Replace non-porous octahedra and dodecahedra with non-porous octahedral- and dodecahedral-shaped particles. It is more appropriate.
4. Lines 27-32: The reference should be placed at the end of the sentence or at relevant place after statement.
below10 nm – Add space.
a consequence of the thermally activated… – as a consequence of the thermally activated…
5. The English should be checked by a native English speaker to enhance the quality of the Manuscript presentation.
6. Line 52: inin vivo – in vivo (with Italic). Pay attention to typos in whole Manuscript.
7. Line 82: (NH2)2CO – (NH2)2CO
8. 2.1. Materials and SCMPIO Synthesis: Add specification for the autoclave you used for the synthesis.
9. If you claim you obtained particles in micrometric range, you must prove it. For example, put the scale on the SEM images or do particle size distribution.
10. You could not claim in the Abstract and Introduction sections that you obtained single crystal structure or pure final Fe3O4 crystal since XRPD confirmed presence of γ-Fe2O3 phase.
11. Line 177: At such low pH without appropriate coating, you risk leakage of iron ions from magnetite structure. This is another reason to avoid this procedure.
12. Maybe, you should reorganize the Manuscript and put 3.5. Hydrothermal Synthesis and chemical reactions at first place in the Results and Discussion section.
Author Response
Reviewer 4
Replace singlecrystalline with single-crystalline.
Response: The change has been made
- The formula Fe(III)-Na4EDTA is incorrect since iron(III) replaces 3Na+ cations from tetrasodium ethylenediaminetetraacetic acid, and so the name Iron-III-Tetrasodium-Ethylenediaminetetraacetic Acid complex is also incorrect. Please, correct it.
Response: We replace “Iron-III-Tetrasodium-Ethylenediaminetetraacetic Acid complex” with “Fe-EDTA complex” and “Fe(III)-Na4EDTA” with “[Fe(III)EDTA]- + Na+”
- Line 16: Replace non-porous octahedra and dodecahedra with non-porous octahedral- and dodecahedral-shaped particles. It is more appropriate.
Response: The change has been made
- Lines 27-32: The reference should be placed at the end of the sentence or at relevant place after statement.
- below10 nm – Add space.
- a consequence of the thermally activated… – as a consequence of the thermally activated…
Response: The change has been made
- The English should be checked by a native English speaker to enhance the quality of the Manuscript presentation.
Response: The English was improved.
- Line 52: inin vivo – in vivo (with Italic). Pay attention to typos in whole Manuscript.
Response: The change has been made.
- Line 82: (NH2)2CO – (NH2)2CO
Response: The change has been made
- 2.1. Materials and SCMPIO Synthesis: Add specification for the autoclave you used for the synthesis.
We completed with: “70 ml teflon - stainless steel line autoclaves”
- If you claim you obtained particles in micrometric range, you must prove it. For example, put the scale on the SEM images or do particle size distribution.
Response: The scale bars was introduced in the SEM images. The suplimentary two SEM images ware added. A histogram was performed and added as figure 1.
- You could not claim in the Abstract and Introduction sections that you obtained single crystal structure or pure final Fe3O4 crystal since XRPD confirmed presence of γ-Fe2O3 phase.
Response: Yes, you are right. We modified the “Abstract” and “Introduction” text.
- Line 177: At such low pH without appropriate coating, you risk leakage of iron ions from magnetite structure. This is another reason to avoid this procedure.
Response: Yes, you are right; that's what happened if the exposure time was longer than three hours; the effects were visible in SEM images. We introduce the “minimizing the risk leakage of iron ions from magnetite structure” in the text.
- Maybe, you should reorganize the Manuscript and put 3.5. Hydrothermal Synthesis and chemical reactions at first place in the Results and Discussion section.
We kept the manuscript in the same configuration.

Round 2
Reviewer 2 Report
The authors have made an effort to address the concrete concerns I had on the submitted manuscript. However, the main issue persists, the content of the proposed manuscript is only the magnetic characterization of the probes, having the preparation, functional characterization and biocompatibility all been reported before by the authors. In my opinion, this content on its own does not justify publication.
Author Response
Response to reviewer 2:
Reviewer’s Comment:
“The authors have made an effort to address the concrete concerns I had on the submitted manuscript”.
Author’s Response: Thank you for the comment.
Reviewer’s Comment:
“However, the main issue persists, the content of the proposed manuscript is only the magnetic characterization of the probes, having the preparation, functional characterization and biocompatibility all been reported before by the authors”.
Author’s Response:
Before submitting our research article for publication to Magnetochemistry, we have carefully read its Aims & Scope message, and we found out that:
- on one hand, the “Aims” section begins with the following phrase:
“Magnetochemistry (ISSN 2312-7481) is an international, scientific open access journal covering all areas of magnetism, from fundamental research on magnetism to applications of magnetic materials, devices, and technologies in all branches of chemistry.”
and
-on the other hand, all the items listed in the “Scope” section:
- Crystal engineering of magnetic materials
- Molecular magnetism
- Magnetic metal–organic frameworks (MOFs)
- Single-molecule, ion, and chain magnets (SMMs, SIMs, and SCMs)
- Spin crossover (SCO) materials
- Magnetic nanostructures
- Magnetic recording
- Magnetocaloric materials
- Qubits
- Theoretical models and calculations
- Applications of magnetic materials
- Magnetic resonances in chemistry
- Magnetic field
Refer to the areas of magnetism mentioned in the “Aims” section.
Accordingly, we consider that our article, in which magnetic characterization (including thermomagnetic analysis) of our magnetite crystals occupies a central placements the aims and scope requirements of this Journal.
However, apart from magnetic characterization, and except from synthesis description and elemental analysis (in our opinion these parts could not be omitted, even if they do not contain essentially novel details compared to Refs. [14], [15], and [16]), morphology characterization, crystal structure analysis, and specific surface and porosity analysis have brought new insights regarding our Fe3O4 microcrystals.
Thus,
- thorough XRD analysis (i.e. long time, and small diffraction angle steps) has revealed the presence of the maghemite, which was attributed to long-term air oxidation; was not reported before (Refs. [14], [15], [16]).
- results on surface and porosity measurements were not reported before (Refs. [14], [15], [16]).
Minor corrections observed by us:
Rows 177-178: Figure. 4 a) The Nitrogen adsorption-desorption isotherm of the magnetite sample b) Pore size Distribution
Row 183: determined using the Brunauer, Emmet, Teller (BET) method, indicating a value of 5,8 m2/g.
Rows 185-188: Applying the Density Functional Theory (DFT) method, an average pore size of approximately 2 nm for Fe3O4 crystallites was obtained. The rugosity (D) of material of 2.0861 determined by the Frenkel, Halsey, and Hill (FHH) method indicates that the material tends to be in a 2 D state, having a
smooth surface [23].
Row 287: where µo is the vacuum permeability, kB is Boltzman’s constant, and mpis the magnetic moment
Row 301-302: HCl, pH = 0.5, 70 °C), minimizing the risk of Fe ions leakage from the magnetite structure or the properties of the magnetite microcrystals. Then, to avoid unwanted FeCO3 formation,
Rows 274-276: was completely eliminated. It was the same thing written twice with rows 277-279.
Academic editor notes: regarding the academic editor notes, we added nine rows (yellow) in the “Conclusions” section.

Reviewer 4 Report
The revised version of the manuscript entitled "Superparamagnetic-like Micrometric Single Crystalline Magnetite for Biomedical Applications- Synthesis and Characterization" meets all criteria to be published in Magnetochemistry journal. The most of the suggestions were taken into consideration and quality of the paper is significantly improved. So, I advise to be accepted.
Author Response
Reviewer’s comment:
The revised version of the manuscript entitled "Superparamagnetic-like Micrometric Single Crystalline Magnetite for Biomedical Applications- Synthesis and Characterization" meets all criteria to be published in Magnetochemistry journal. The most of the suggestions were taken into consideration and quality of the paper is significantly improved. So, I advise to be accepted.
Response: Thank you for the comment.
Minor corrections observed by us:
Rows 177-178: Figure. 4 a) The Nitrogen adsorption-desorption isotherm of the magnetite sample b) Pore size Distribution
Row 183: determined using the Brunauer, Emmet, Teller (BET) method, indicating a value of 5,8 m2/g.
Rows 185-188: Applying the Density Functional Theory (DFT) method, an average pore size of approximately 2 nm for Fe3O4 crystallites was obtained. The rugosity (D) of material of 2.0861 determined by the Frenkel, Halsey, and Hill (FHH) method indicates that the material tends to be in a 2 D state, having a
smooth surface [23].
Row 287: where µo is the vacuum permeability, kB is Boltzman’s constant, and mpis the magnetic moment….
Row 301-302: HCl, pH = 0.5, 70 °C), minimizing the risk of Fe ions leakage from the magnetite structure or the properties of the magnetite microcrystals. Then, to avoid unwanted FeCO3 formation….
Rows 274-276: was completely eliminated. It was the same thing written twice with rows 277-279.
Academic editor notes: regarding the academic editor notes, we added nine rows (yellow) in the “Conclusions” section.
